# GOODFIT: A DEEP LEARNING OPTIMIZER FINE-TUNED FOR FINE TUNING

## ABSTRACT

We present GOODFIT, one of the first optimizers that has been specifically designed to operate on converged models that need to be incrementally fine-tuned on a new task/dataset. Unlike standard optimizers like SGD or Adam, which operate with minimal assumptions since the model weights might be randomly initialized, GOODFIT takes advantage of the additional structure of a converged model to regularize the optimization process for better results. GOODFIT uses a simple temporal gradient orthogonalization process to outperform traditional fine-tuning methods in a wide variety of settings, from long-tailed classification to large-scale motion prediction. And because GOODFIT is fully encapsulated within the logic of an optimizer, it can be trivially dropped into any model training pipeline with minimal engineering effort. We believe that a new class of fine-tuning optimizers like GOODFIT can help pave the way as fine-tuning and incremental training become more and more prevalent within modern deep learning, and practitioners move further and further away from expensively training models from scratch.

## 1 INTRODUCTION

Training large neural network models from scratch is expensive. As datasets and models increase in size, having to train a new model for every new setting quickly becomes intractable. Imagine, for example, the cost of having to train a new model every time an autonomous vehicle needs to operate in a new city. The shift towards fine-tuning within the deep learning community has also been accelerated by developments in large foundational models that were trained on vast quantities of data, such as Large Language Models (LLMs)Brown et al. (2020). Indeed, hints abound that deep learning is steadily inching towards a new paradigm where only simple models are trained from scratch.

At the same time, fine-tuning a model comes with its own set of challenges. For one, it is well known that models tend to readily forget old information when fine-tuned on new information, in a process known as "catastrophic forgetting." Various mitigation methods have been proposed Li & Hoiem (2017), but often require extensive additional data engineering and modifications of the model architecture. The common practice within fine-tuning still seems to largely rely on training on new tasks/data with a smaller learning rate or with a frozen backbone (or both). The difficulty of developing widely-adopted fine-tuning methods often lies in the requisite generality and simplicity of such a method. We aim to address both of these desirable properties in the present work.

Speaking of generality and simplicity, one ubiquitous element within deep learning training that is also almost always implemented in a modular fashion is the optimizer. The majority of models set the optimizer to some noncontroversial choice (such as Adam Kingma & Ba (2015); Loshchilov & Hutter (2019) or Momentum Sutskever et al. (2013)). Yet, all current popular optimizers are designed with training-from-scratch in mind, which forces their design to assume minimal structure about the problem setting. In contrast, the fine-tuning setting usually starts from a well-trained, well-converged model that we already trust. So we ask: *how do we design an optimizer especially for the case where we start from a converged model*? Given the ubiquity and modularity of optimizers, such an optimizer would be immediately applicable and easy to implement within any fine-tuning setting.

Works such as Learning Without Forgetting (LWF) Li & Hoiem (2017) have proposed mitigating fine-tuning regression by sticking close to an old state of the model, but requires additional data pipelining and model snapshots that serve as additional supervision to keep the model anchored to its old "good" state. Instead of anchoring a model across tasks, we propose a slightly different but

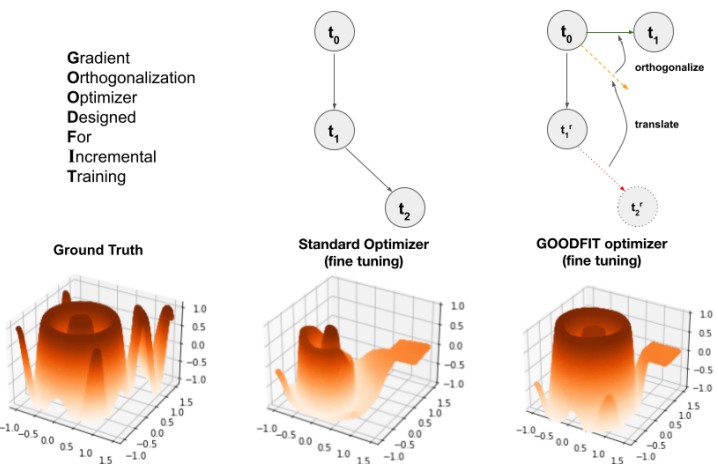

Figure 1: Schematic of a gradient update for GOODFIT. Unlike standard optimizers, which take successive steps through parameter space, GOODFIT keeps track of a "good" model state and collects reference information of how the system tries to move away from the "good" state (denoted by the superscript "r" - for "reference" - in the figure). The functionality of the "good" state is then restored by translating and orthogonalizing gradient updates with respect to that state. These processes allow GOODFIT to take heavily regularized update steps and produce superior performance when in challenging fine-tuning settings. The visualization in the bottom row is discussed in Section 4.1.

equally valid anchoring *across time*. For each iteration of the optimizer, we can take a step away from equilibrium and *balance further steps away from equilibrium with the model's desire to return to equilibrium* using standard methods from the gradient-based multitask learning literature. The result is a system that mimics data-driven anchor methods such as Li & Hoiem (2017) without actually needing data and replaces the rigid anchors with a dynamically updated flexible one.

Specifically, a model converged in some state $\mathbf{x}_0$ will proceed along the states $\mathbf{x}_1, \dots \mathbf{x}_t$ with a standard optimizer. A further update would send the system to $\mathbf{x}_{t+1}$. However, because $\mathbf{x}_0$ was a "good state," the model also would benefit by returning to $\mathbf{x}_0$. We thus have two potentially conflicting gradient directions ($\boldsymbol{\Delta} := \mathbf{x}_t \to \mathbf{x}_0$ and $\mathbf{g} := \mathbf{x}_t \to \mathbf{x}_{t+1}$), which is a classic multitask learning problem. We borrow some ideas from Yu et al. (2020) and assign $\mathbf{g} \mapsto \mathbf{g} \perp \boldsymbol{\Delta}$, where $\perp$ is the orthogonalization operator. We then restore the model to state $\mathbf{x}_0$ (the "translate" operation on the top right of Figure 1) and take a step in the orthogonalized $\mathbf{g}$ direction. We name our method GOODFIT, which stands for Gradient Orthogonalization Optimizer Designed for Incremental Training, and a schematic of the process is shown in Figure 1. To the best of our knowledge, GOODFIT is one of the first optimizers that is specifically designed to work in the fine-tuning/incremental training setting.

Note that the mathematical operation of GOODFIT depends on the assumed compatibility of the fine-tune setting with the baseline setting, as GOODFIT tries to dynamically keep the model state close to that of the baseline. GOODFIT excels at gracefully adapting a model towards a *nearby distribution* of interest, and this compatibility is therefore a core element of the setting we tackle. While this may exclude some use cases that practitioners may refer to as "fine-tuning" (e.g. pretrain on ImageNet, fine-tune on an unrelated dataset), we wish to be careful in separating those pretraining settings with the majority of practical fine-tuning applications that fall under the compatibility constraint. GOODFIT provides significant performance boosts in those cases, as we will demonstrate.

Our main contributions are as follows:

- We present GOODFIT, the first optimizer designed specifically for fine-tuning a good baseline model, and which can be easily dropped into any deep learning training framework.

- We theoretically prove that GOODFIT allows us to train as if we still had access to the old data/setting in an unsupervised way.

- We experimentally show that GOODFIT beats standard fine-tuning methods in a large number of settings, from a low-dimensional toy example to image classification to large-scale motion prediction for autonomous driving.

## 2 RELATED WORK

**Fine-Tuning** Fine-tuning a pre-trained model on particular datasets Kornblith et al. (2019); Chen et al. (2020) is a common technique in the era of deep learning. For fine-tuning purposes, practitioners often use standard popular optimizers like Stochastic Gradient Descent (SGD) Bottou (2010) and AdamW Kingma & Ba (2015); Loshchilov & Hutter (2019). Recent advanced modern architectures such as ViTs Dosovitskiy et al. (2021); Caron et al. (2021); Radford et al. (2021) or ConvNeXts Liu et al. (2022) use AdamW for fine-tuning, while it is also common to use SGD for fine-tuning models like ResNets He et al. (2016); Kolesnikov et al. (2020) due to the optimizer's efficiency. However, SGD and AdamW do not assume that we want to stay close to our model's start state, and thus lead to models that tend to forget old data when fine-tuning on new data, also known as catastrophic forgetting or catastrophic interference McCloskey & Cohen (1989). Our method serves as a regularization approach to bridge the gap in the existing optimizers to mitigate this forgetting issue.

**Continual Learning** To address this issue, various approaches have been developed, such as regularization-based methods Kirkpatrick et al. (2017); Chaudhry et al. (2020); Jung et al. (2020); Titsias et al. (2020); Iman Mirzadeh et al. (2021), where the goal is to keep learned information of the past tasks during continual learning. In particular, Mirzadeh et al. Iman Mirzadeh et al. (2021) demonstrated a link between continual learning and multi-task learning, and Farajtabar et al. Farajtabar et al. (2020) proposed projecting gradients from new tasks onto the subspace of prior task gradients. Learning Without Forgetting (LWF) Li & Hoiem (2017); Masana et al. (2022) proposes storing the old model response on new tasks/data for additional supervision. Our method draws parallels with these approaches but does not rely on storing costly amounts of old data/statistics, which makes our method much more plug-and-play and dynamic. Two other directions to address the catastrophic forgetting are rehearsal-based methods Rebuffi et al. (2017); Chaudhry et al. (2019a); Lopez-Paz & Ranzato (2017); Chaudhry et al. (2019b); Saha et al. (2022) that directly make use of the old data source, and architecture-based methods that minimize the inter-task interference via new architectures Mallya & Lazebnik (2018); Serra et al. (2018); Li et al. (2019); Wortsman et al. (2020); Wu et al. (2019). These methods also generally add substantial infrastructural overhead, while our approach is attractive in its simplicity of implementation.

**Multi-Task Learning** For detailed background context in multitask learning, we refer the reader to Zhang & Yang (2021). MTL Zhang & Yang (2021) is an optimization problem where we concurrently train a model on multiple tasks to take advantage of the structures of shared neural networks, thus improving generalization. One MTL direction is to use gradient descent method to optimize joint multi-task learning. Our problem setting shares some similarities with that. The difference is that our problem is considered temporal multi-task learning. In MTL literature, Ozan Sener & Koltun (2018) formulates the MTL problem as a multi-objective optimization problem and then learns the loss weights that change dynamically. GradNorm Chen et al. (2018) tries to normalize the gradients to balance the learning of multiple tasks. PCGrad Yu et al. (2020) suggests that to mitigate issues with gradient direction conflicts, we should project a task's gradient onto the normal plane of the gradient of any other tasks where a gradient conflict is present. Our proposed technique modifies some of the core PCGrad designs specifically for the fine-tuning setting.

## 3 METHODOLOGY

We now describe the complete GOODFIT algorithm, along with a discussion of hyperparameters and implementation details. At a high level, GOODFIT consists of two separate standard optimizers, $\mathbf{O}$ and $\mathbf{O}^{(\text{ref})}$. The latter "reference" optimizer perturbs the system from equilibrium, while the former "main" optimizer uses this perturbation to make a final update. The entire logic of GOODFIT can be encapsulated within the logic of this single optimizer class, which makes the method very portable and modular. We finish with a few theoretical considerations relevant to GOODFIT.

## 3.1 THE GOODFIT OPTIMIZER

The GOODFIT optimizer is exceedingly easy to understand and implement. All its logic can be easily encapsulated within the definition of an optimizer step, and it relies on no mathematics more complicated than vector orthogonal projections. We describe GOODFIT in Alg 1.

---

**Algorithm 1** The GOODFIT fine-tuning optimizer

---

**Require:** Converged model $\mathcal{M}(\mathbf{x}; \theta)$ to be trained on data $\mathbf{X}'$ with loss $L$.
**Require:** Initialize reference model weights $\theta_{\text{ref}}$.
**Require:** Initialize batch size $B$, reference steps $n_{\text{ref}}$, training steps $n_{\text{steps}}$, standard optimizer $\mathbf{O}$ with learning rate $\lambda_{\text{main}}$, and reference optimizer $\mathbf{O}^{(\text{ref})}$ with learning rate $\lambda_{\text{ref}}$. Each optimizer takes as arguments the current weights and a gradient update direction, producing updated weight values.

1: **for** $n_{\text{step}}$ steps **do**:
2:     $\theta_{\text{ref}} \leftarrow \theta$                         ▷ Save the model state.
3:     **for** $n_{\text{ref}}$ steps **do**
4:         Take new $B$ examples from $\mathbf{X}'$ and calculate gradients $\mathbf{g} := \nabla_\theta L$.
5:         Take one step with reference optimizer $\theta \leftarrow \mathbf{O}^{(\text{ref})}(\theta, \mathbf{g})$.
6:     **end for**

7:     Calculate $\Delta = \theta - \theta_{\text{ref}}$.     ▷ Calculate total displacement during reference steps.
8:     Find $\mathbf{g} := \nabla_\theta L$ for a new batch as in Line 4.
9:     Calculate dot product $\omega = \langle \Delta, \mathbf{g} \rangle$.
10:    **if** $\omega < 0$ **then**:
11:        $\mathbf{g} \leftarrow \mathbf{g} \perp \Delta$         ▷ $\mathbf{a} \perp \mathbf{b}$ denotes orthogonal projection of $\mathbf{a}$ onto $\mathbf{b}$
12:    **end if**
13:    $\theta \leftarrow \theta_{\text{ref}}$.                   ▷ Restore original state.
14:    $\theta \leftarrow \mathbf{O}(\theta, \mathbf{g})$            ▷ Take step with main optimizer.
15: **end for**

---

Given a model $\mathcal{M}$ parameterized by weights $\theta$ which has been trained on data $\mathbf{x}$ corresponding to an upstream task, and a data source $\mathbf{X}'$ corresponding to a new task (with a task loss $L$), we aim to fine-tune $\mathcal{M}$ to work well on both the old and new tasks. Note that we make a strict assumption about having a converged model $\mathcal{M}$ available as an input to GOODFIT; any random or untrained weights in the model can lead to poor performance. Additionally, our model also requires two instantiated optimizers, a standard optimizer $\mathbf{O}$ and a reference optimizer $\mathbf{O}^{(\text{ref})}$.

At each step of training, we first store the current state of $\mathcal{M}$ by saving $\theta$ into $\theta_{\text{ref}}$. Next, we sequentially draw $n_{\text{ref}}$ batches from $\mathbf{X}'$ and iteratively minimize $L$ with the reference optimizer $\mathbf{O}^{(\text{ref})}$. We have now perturbed the system from equilibrium and must re-establish said equilibrium.

To do so, we first calculate $\Delta := \theta - \theta_{\text{ref}}$, the total displacement of our original position after $n_{\text{ref}}$ steps of the optimizer $\mathbf{O}^{(\text{ref})}$. We reason that if the stored equilibrium state corresponds to a good critical point of the original model, then $\Delta$ corresponds to the benign gradient direction that will restore the original critical point. We thus have two potentially conflicting gradient updates: the one corresponding to $\Delta$, and the one corresponding to the next queried update by $\mathbf{O}^{(\text{ref})}$, which we call $\mathbf{g}$. We need to decide on how to take a single gradient step that is consistent with both of these options.

We borrow an idea from PCGrad Yu et al. (2020), which reconciled conflicting gradients by orthogonally projecting them onto each other in a pairwise fashion. Crucially, we choose to only project $\mathbf{g}$ onto $\Delta$, and not the other way around, because $\Delta$ represents a gradient on the old dataset which may no longer be accessible and, therefore, must be treated with more care. Therefore, we end with the two gradient updates $\mathbf{g}$ and $\mathbf{g} \perp \Delta$, and we take both steps by first restoring $\theta \mapsto \theta_{\text{ref}}$ and then allowing $\mathbf{O}$ to take a step in the $\mathbf{g} \perp \Delta$ direction. This process is repeated until training is complete.

## 3.2 HYPERPARAMETER DISCUSSION AND TRADEOFFS

GOODFIT introduces three main hyperparameters: $n_{\text{ref}}$, $\mathbf{O}^{(\text{ref})}$, and $\lambda_{\text{ref}}$. As mentioned, higher $n_{\text{ref}}$ allows $\mathbf{O}^{(\text{ref})}$ more flexibility to explore, while $\lambda_{\text{ref}}$ controls the step size at each iteration of the

reference step. The choice of $\mathbf{O}$ is flexible, but $\mathbf{O}^{(ref)}$ should generally be set to standard SGD, as the gradient calculations that drive GOODFIT are cleanest when the reference updates are simple.

The main practical concern of GOODFIT would be the cost of setting $n_{\text{ref}}$, as our model must take $n_{\text{ref}}$ additional optimization steps per training step. In practice, $n_{\text{ref}} = 1$ will often work well, and the practitioner should only increase $n_{\text{ref}}$ if they observe poor performance. Even then, the potentially increased compute of GOODFIT is justifiable: (1) we find that GOODFIT generally converges quicker, possibly due to better regularization effects, and (2) fine-tuning conventionally runs for fewer steps than from-scratch training, which minimizes the impact of additional training time.

Mathematically, $\lambda_{\text{ref}}$ encodes how quickly we travel from equilibrium to study the general shape of the loss surface, so that we can make an informed decision on where to go next. In practice, we find that there is often a sweet spot somewhere between $\lambda_{main}/100$ and $\lambda_{main}/10000$, but the exact value is heavily dependent on the loss surface for the specific problem. It may in fact turn out that the optimal value of $\lambda_{\text{ref}}$ tells us something about the fundamental properties of that particular setting's loss surface, but such analysis falls outside the scope of the present work.

### 3.3 THEORETICAL CONSIDERATIONS

We now list a few simple theoretical properties of GOODFIT, along with sketches of proofs. We first prove that GOODFIT is "correct," in the sense that it is able to decrease the loss on the old model task/setting. We also add a discussion of when GOODFIT might fail by enumerating all stable points of GOODFIT, but argue why this is generally not an issue in practical settings.

(**Correctness on old data.**) Take a model $\mathcal{M}(\mathbf{x}; \theta)$ converged on data $\mathcal{X}_{\text{old}}$ at a local minimum of loss $L_{\text{old}}$. We would now like to fine-tune on data $\mathcal{X}_{\text{new}}$ with loss $L_{\text{new}}$. Suppose that $\mathbf{O}^{(\text{ref})}$ takes $\theta \mapsto \theta'$. A single step of GOODFIT on batch $\mathbf{x}_{\text{new}}$ with a sufficiently small learning rates $\lambda_{\text{main}}, \lambda_{\text{ref}}$ will decrease $L_{\text{old}}(\mathcal{X}_{\text{old}})$ from its value at $\theta'$.

Proof Sketch: If $\mathcal{M}$ is converged on data $\mathcal{X}_{\text{old}}$, then it must be at a local minimum of the old data. If the learning rate is sufficiently small, then $\nabla_{\theta'} L_{\text{old}}$ must point in the $\Delta$ (as defined in Alg 1) direction, and so an update with the gradient $\Delta + \langle \Delta, \mathbf{g} \rangle$ will necessarily decrease $L_{\text{old}}$. $\square$

The prior theorem establishes that GOODFIT accomplishes precisely what it seeks to do: even if the old data is no longer available, GOODFIT allows us to train as if we can still compute the full loss function of the old data! As the system moves further away from the old equilibrium, we are able to restore some of the function of that equilibrium through these regularized updates. However, it is important to note an important case where GOODFIT results in a trivial update.

(**Stable points.**) Suppose a model has weights $\theta$ and $\mathbf{O}^{(\text{ref})}$ maps $\theta \mapsto \theta'$ and $\mathbf{O}^{(ref)}$ is SGD. If we are not at a critical point of $L_{\text{new}}$, GOODFIT will result in zero change in the model weights $\mathbf{w}$ if and only if $\hat{\nabla}_\theta L_{\text{new}} = \hat{\nabla}_{\theta'} L_{\text{new}}$, where $\hat{\nabla}$ refers to the unit vector corresponding to $\nabla$.

Proof: $\Delta$ in this case would point in the direction $-\nabla_\theta L_{\text{new}}$, at which point $\nabla_\theta L_{\text{new}} \perp \Delta = 0$, and the total update from GOODFIT will be just a restoration to $\theta$. If the gradient equality does not hold, $\nabla_\theta L_{\text{new}} \perp \Delta \neq 0$ will lead to a non-zero total update. $\square$

**Linearity forces a stable point.**) In the situation defined by Theorem 3.3, GOODFIT will encounter a stable point if the loss surface is perfectly linear between $\theta$ and $\theta'$. $\square$

The prior theorem and corollary demonstrate that GOODFIT will fail to move the system when the loss surface becomes exactly locally linear at a point. Luckily, this almost never occurs for high-dimensional loss surfaces that exist for deep models, and even in the 2-dimensional toy example described in Section 4.1 we see a reasonable performance.

## 4 EXPERIMENTS

We now detail a number of experiments for GOODFIT in diverse settings, from a low-dimensional toy setting to image classification to large-scale motion prediction. We primarily focus on comparisons to standard fine-tuning with standard optimizers (on either the full model or just the model head), as those are - by a large margin - still the most commonly used fine-tuning methods in the industry due

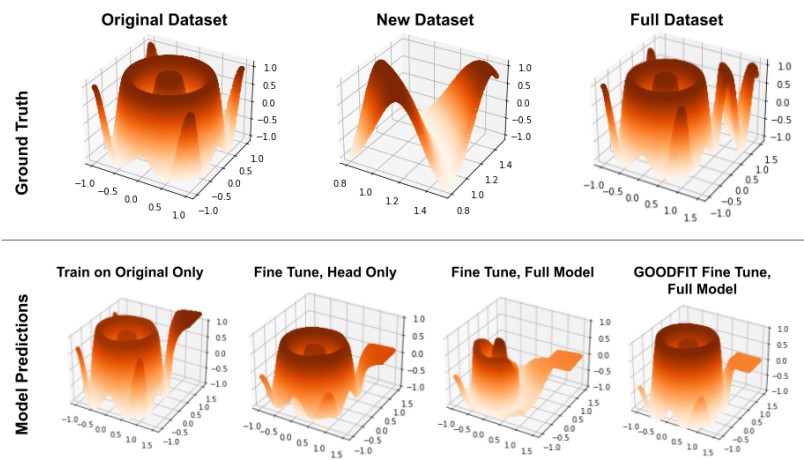

Figure 2: Fine-tuning toy example visualizations.

to their known performance and ease of implementation. We will show that GOODFIT, while being just as easy to implement, provides a significant performance boost in all cases.

### 4.1 A SIMPLE TOY EXAMPLE

We present results in this section on a simple, low-dimensional toy example. We find that testing new algorithms in low-dimensional settings is always instructive and provides us with insight that cannot be gleaned from the noise of higher-dimensional experiments.

Namely, we fit a number of simple MLP models onto a 2D function. We pick the 2D function $f(\mathbf{x}) = \sin(10|\mathbf{x}|)$, because both radially symmetric and periodic functions are generally more challenging for neural networks to fit. We further add $\mathcal{N}(0, 1)$ noise to the output. Even though we want to fit to a low-dimensional example, it is still important for the problem to be difficult enough to see interesting behavior within our models, especially given the dimensionality requirements as discussed in Section 3.3. The original dataset consists of input-output pairs where input coordinates are drawn independently from $\mathcal{U}[-1.0, 1.0]$, while the new dataset we wish to fine-tune on has input coordinates drawn from $\mathcal{U}[0.8, 1.5]$. This is clearly also challenging because, although the domains of the two datasets overlap in the interval $[0.8, 1.0]$, they are largely nonoverlapping.

Our MLP consists of three layers with weights of shape $[2, 500] \rightarrow [500, 500] \rightarrow [500, 1]$, and we use RMSProp as our baseline optimizer. The baseline model is trained on the original data split only for 10000 steps at a learning rate of 1e-2, with fine-tuning runs trained at a learning rate of 5e-4 for 1500 steps. For exact details on the training procedure, please refer to the supplementary material.

| Method | Original Data Error ($\downarrow$) | New Data Error ($\downarrow$) |
|---|---|---|
| Baseline Trained on Original Data | 0.0054 | 1.907 |
| fine-tune on New Data (Full Model) | 0.705 | 0.504 |
| fine-tune on New Data (Head Only) | 0.110 | 0.572 |
| GOODFIT on New Data | **0.046** | **0.501** |

Table 1: Results on a 2D fitting problem. A baseline is trained on just the original data domain and then fine-tuned at a lower learning rate on a new data domain (both the full model and the head only). This is compared to GOODFIT, which is trained on the full model. As expected, there is a clear tradeoff between training on the full model versus the head only; the latter provides some protection against performance regression on the original data, but also is less able to adapt to the new data. GOODFIT outperforms both baselines and shows impressive resilience in maintaining performance on the original split. All results have standard error $\leq 0.01$.

The results are visualized in Figure 2. Visually, the benefits of GOODFIT are pronounced; fine-tuning on the full model results in significant warping, and even fine-tuning on only the head layer still produces undesirable shifts in the height of the output shape. GOODFIT effectively remembers the original shape with only minimal regressions along the steep edges of the distribution.

Numerically, as shown in Table 1, GOODFIT outperforms the baseline in not forgetting the original dataset distribution. Even though GOODFIT modifies all model weights, it mitigates forgetting relative to the head-only fine-tuning baseline by a sizable margin, even though GOODFIT acts on significantly more weights. Fine-tuning of the full model in a naïve way weights leads to disastrous results, with significant deformations of the predictions. GOODFIT allows us the flexibility of full-model fine-tuning without the drawbacks of severe catastrophic forgetting.

Though GOODFIT is partially inspired by LWF Li & Hoiem (2017), it is worth noting that LWF is ill-suited for shifting dataset domains, as it uses evaluation on the new data with the old model to generate additional supervision. With large input domains shifts, this supervision would be of poor quality. Thus, these experiments also serve to show that GOODFIT is a quite general fine-tuning tool.

## 4.2 Long-Tailed Image Classification

Next, we test our method on a challenging long-tailed image classification problem. For these experiments, we work with the CIFAR100-LT Krizhevsky et al. (2009); Cui et al. (2019) dataset. The frequencies per class are controlled using a smooth exponential decay function and the imbalance ratio $\beta$, which corresponds to the frequency between the most and least frequent classes.

Our baseline is GLMC Du et al. (2023) using SGD with Momentum, a method which offers state-of-the-art performance on CIFAR100-LT (we confirmed 72.89% accuracy with $\beta = 10$). GLMC uses a ResNet-34 backbone and involves a single-stage training procedure. A contrastive consistency loss is applied between these batches for robust feature extraction, in addition to classification losses for each batch. For further details, we refer the reader to the supplementary material.

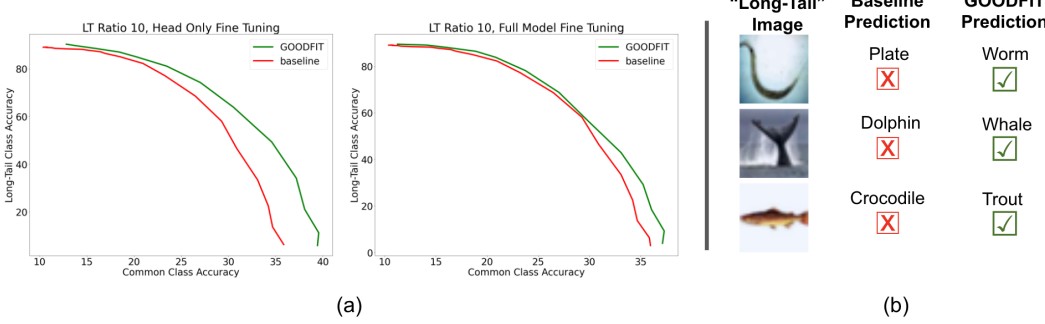

Figure 3: CIFAR100-LT Results. (a) Trade-off curves for $\beta = 10$ for performance on the base versus the long-tailed classes. This is the most imbalanced data setting we tested and clearly demonstrates the better performance of GOODFIT. (b) Some sample visualizations of images in the long-tailed classes and their associated prediction outputs for GOODFIT versus the baseline fine-tuned model.

To test GOODFIT, we add a fine-tuning second stage to the training pipeline. In the first stage, we train the networks with classes $[90, 99]$ held out. Using this initial state, the fine-tuning stage involves training the network only on the held-out classes $[90, 99]$. We test on imbalance ratios $\beta = [2, 5, 10]$.

Our results are detailed in Figure 3 and Table 2. In general, by fine-tuning only on long-tailed data, we trace out a sharp trade-off in performance on the base and long-tailed classes. The tradeoff shows that the model has the tendency to catastrophically forget its knowledge of the base classes, making this setting a perfect test for GOODFIT. In all cases and all ratios, the model fine-tuned by GOODFIT outperforms those of the standard fine-tuning methods. The improvement is most pronounced for cases of extreme imbalance ($\beta = 10$), as seen in the tradeoff curves displayed to the left of Figure 3.

Fine-tuning the classification head outperforms full model fine-tuning for LT Acc @ 30 with $\beta = 5$. However, fine-tuning on the head only is particularly favorable within this setting, as freezing model

| Imbalance | Method | Weights | LT Acc @ 40 (↑) | LT Acc @ 35 (↑) | LT Acc @ 30 (↑) |
|---|---|---|---|---|---|
| 2 | Baseline | Head Only | 16.99 | 35.62 | 54.26 |
| | GOODFIT | Head Only | **34.69** | **48.06** | **61.45** |
| 2 | Baseline | Full Model | 27.49 | 38.24 | 49.19 |
| | GOODFIT | Full Model | **29.54** | **43.76** | **57.98** |
| 5 | Baseline | Head Only | 20.02 | 35.19 | 50.36 |
| | GOODFIT | Head Only | **24.22** | **37.43** | **50.64** |
| 5 | Baseline | Full Model | 20.89 | 33.97 | 47.03 |
| | GOODFIT | Full Model | **24.36** | 35.72 | 47.08 |
| 10 | Baseline | Head Only | 9.56 | 24.52 | 39.48 |
| | GOODFIT | Head Only | **18.19** | **33.33** | **48.47** |
| 10 | Baseline | Full Model | 6.47 | 22.05 | 37.63 |
| | GOODFIT | Full Model | **19.5** | **30.14** | **40.78** |

Table 2: Results on CIFAR100-LT. The standard error is within 0.01%. Due to the trade-off between the performance on the base and long-tailed class, we evaluate the baselines and GOODFIT performance on the long-tail class when the base class accuracy is 40%, 35%, and 30% respectively.

weights provides insulation from the severeness of forgetting. In fact, applying GOODFIT to the head weights only provided an even further performance boost. Thus, although we generally recommend applying GOODFIT to all model weights, in extreme incidents GOODFIT can be effective even when applied to a subset of weights alone.

### 4.3 LARGE-SCALE ROBOTICS MOTION PREDICTION

We now go to the other end of the spectrum and show how GOODFIT performs on an extremely challenging benchmark with a large, high-dimensional dataset. The Waymo Open Motion Dataset (WOMD) Ettinger et al. (2021) is a large-scale driving dataset collected from realistic scenarios. The task is to predict future trajectories of an agent over the next 8 seconds, given multi-modal observations in the last second, including the agent's history, nearby agent histories, map information, and traffic light states. The prediction model follows a state-of-the-art early-fusion transformer architecture Nayakanti et al. (2022), by fusing multi-modal input features through a self-attention transformer and predicting future trajectory samples using learned latent queries.

We first train a WOMD model on car trajectory prediction, and then fine-tune that model on data from the same class (car) as well as different classes (pedestrian). The WOMD is not just a large-scale version of the CIFAR classification setting described in Section 4.2, but offers an important difference in that we can see how well GOODFIT performs on (1) fine-tuning a model on the same exact data, and (2) fine-tuning a model on a completely different domain-shifted task. The long-tailed data in CIFAR100 was still explicitly part of the same distribution as the other data, but pedestrian trajectories have semantically different behavior than car trajectories.

The results are shown in Figure 4 and Table 3. Average Distance Error (ADE) measures the average distance between ground truth and prediction at each point of the predicted trajectories, while Final Distance Error (FDE) only considers the error in the final point of the predicted trajectories. We can see that in all cases GOODFIT outperforms the baseline fine-tuning methods, with sizable benefits in the car-to-pedestrian fine-tuning task. Consistent with the results in other settings, fine-tuning with head only provides subpar performance on the car-to-pedestrian task, while full model fine-tuning does reasonably well but is still outstripped by GOODFIT. We conclude that GOODFIT provides a superior ability to repurpose a model for a new, correlated setting.

Interestingly, we also see minor but noticeable improvements in the car-to-car benchmark, suggesting that GOODFIT can be used to extract more performance from any model that has already converged. We could even imagine a scenario in which regular additional training runs using GOODFIT are enabled as a standard model maintenance practice.

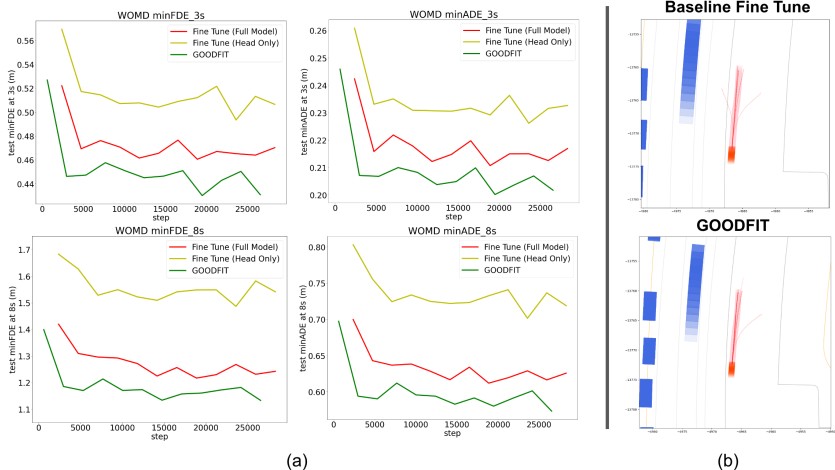

(a)                                                                                          (b)

Figure 4: Results for GOODFIT on Waymo Open Motion Dataset. (a) Error curves through training tabulated for FDE at 3s and 8s (top) and ADE at 3s and 8s (bottom). GOODFIT outperforms both fine-tuning baselines by a sizable margin. (b) Visualizations of motion prediction outputs for both the baseline fine-tune model (top) and GOODFIT (bottom). Trajectory ground truth is shown as a shaded bar and denser lines represent more confident predictions. Although the differences are often subtle, GOODFIT generally produces more confidently correct predictions.

| Method | Target Class | ADE@3s (m) | ADE@8s (m) | FDE@3s | FDE@8s |
|---|---|---|---|---|---|
| Baseline | - | 0.461 | 1.327 | 1.024 | 2.581 |
| fine-tune (F) | Car | 0.458 | 1.322 | 1.021 | 2.548 |
| fine-tune (H) | Car | 0.456 | 1.303 | 1.009 | 2.507 |
| GOODFIT | Car | **0.454** | **1.299** | **1.008** | **2.489** |
| fine-tune (F) | Ped | 0.214 | 0.621 | 0.465 | 1.242 |
| fine-tune (H) | Ped | 0.232 | 0.724 | 0.508 | 1.544 |
| GOODFIT | Ped | **0.203** | **0.579** | **0.427** | **1.145** |

Table 3: Results on Waymo Open Motion Dataset. F stands for full-model fine-tuning and H stands for head only. Standard errors are within 0.005m for 3s metrics and 0.015m for 8s metrics. There is a minor but noticeable improvement for GOODFIT on the car-to-car benchmarks and a substantial improvement for GOODFIT on the car-to-ped benchmarks.

## 5 CONCLUSION

We proposed GOODFIT, an optimizer that is specifically designed to robustify models during fine-tuning. Unlike standard optimizers which cannot assume much about a training setting, our ability to assume confidence about the prior state of a model allows GOODFIT to act as an effective regularizer and prevent a model's weights from diverging too much from its previously good state. We showed that GOODFIT performs well in various settings: (1) fine-tuning to new data off the data manifold in our toy example, (2) fine-tuning to new data from new classes on CIFAR100-LT, and (3) fine-tuning a model on a completely new task as well as on the same exact task on large-scale motion prediction. In all cases, GOODFIT surpassed the performance of the near-ubiquitously used, standard fine-tuning approaches, while being just as easy to integrate within a generic model training pipeline. We believe that GOODFIT should not only be an important new tool in the deep learning practitioner's arsenal but opens the discussion around developing a new class of optimizers that will crucially support the new age of deep learning where fine-tuning becomes the primary training paradigm.

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
