# SUPPLEMENTARY MATERIAL FOR
# GOODFIT: A DEEP LEARNING OPTIMIZER FINE-TUNED FOR FINE TUNING

## ABSTRACT

In the supplementary material, we provide the following:

- Additional motivation regarding GOODFIT (Sec. 1).
- A theoretical discussion about GOODFIT (Sec. 2).
- Detailed setup and hyper-parameters for baseline and fine-tuning experiments (Sec. 3).
- The limitations of using GOODFIT (Sec. 4).

## 1 MOTIVATION

In practical deep learning use-cases, a network constantly needs to be updated using new data or tasks. For instance, consider the case of an autonomous vehicle. As the agent operates across numerous cities, it gathers different types of sensor data unique to each city. To re-train the network from scratch each time quickly becomes intractable due to several reasons: (a) old data needs to be retained (i.e., storage costs), (b) requires extensive compute to re-train, and (c) discards validation effort required for the "previous" model. One potential approach to overcome these issues is to freeze the backbone, while only fine-tuning the relevant fully-connected classification layers. However, this approach assumes that the backbone is capable of providing meaningful features, even for data from a new domain (e.g., a new city), something which CNNs have notoriously struggled with.

While Learning Without Forgetting (LWF) Li & Hoiem (2017) aims to address some of these challenges, it still suffers from some key challenges: (a) hyper-parameter tuning to balance old and new task loss weights, (b) access to the previous model checkpoint, and (c) requires two model forward passes in each iteration which may be computationally expensive. In order to address all these challenges in literature, our key insight is that the optimization objective in previous approaches does not leverage the structure of a well-initialized pretrained model. Concretely, we treat the pretrained weights as a location on the optimization landscape where a random initialized model may desire to be to get good performance on the old task.

## 2 ADDITIONAL THEORETICAL DISCUSSION

Here, we provide a bit more exposition on the theoretical properties of GOODFIT. In the main text (Section 3), we proposed two properties of GOODFIT: (1) that GOODFIT updates will reduce the loss value for the old loss on the old data, despite making no assumptions on access to the old data, and (2) that GOODFIT has stable points on linear loss surfaces.

The implications of statement (1) are fairly straightforward, as it implies that we can train with the settings of the old system even when that old system falls out of scope. This is the primary feature of GOODFIT, and is the main proof that GOODFIT is a meaningful regularization method. But the implications of (2) are more interesting. The fact that GOODFIT works despite not functioning properly in linear loss surfaces implies that GOODFIT relies on nontrivial values of second-order gradients and curvature within the loss surface to function. Thus, any critical point that the system converges to while under GOODFIT updates must have been reached through a nonlinear path from the model's starting point. This requirement may have robustness implications for the convergence

points found by GOODFIT, as any such convergence points must have alternative nonlinear paths leading to it.

There is an important discussion to be had though regarding the convergence of GOODFIT. We did not expand on this convergence in the main text because talking about convergence within a fine-tuning setting is somewhat tricky at a fundamental level, as it is difficult to deconvolve convergence on the new dataset with convergence on the old setting (for which we may no longer have available data). Especially when the number of incremental training stages increases, it becomes secondary to converge on the new (potentially small) dataset that the model is fit on and more important to maintain the efficacy of the base model. In that case, the correctness property (1) becomes the main property of importance.

In general, a proof of convergence of GOODFIT is difficult for two reasons: (1) The orthogonalization procedure we use is asymmetric, versus the symmetric procedure in Yu et al. (2020), due to our prioritization of mitigating regressions, and (2) the restoration step to the old state after steps of $\mathbf{O}^{(\mathrm{ref})}$ is discrete rather than treated as a separate incremental gradient step. Both of these design decisions were made to support the main goal of regression mitigation for applications of GOODFIT. So although we do not provide a complete proof of conditions under which GOODFIT converges in this work, we make this omission precisely because convergence on the new data is a secondary goal in our setting, and the majority of our important design decisions were not made in service to that particular goal, but rather to the more difficult goal of regularization during fine-tuning.

## 3 EXPERIMENTS

### 3.1 A TOY EXAMPLE

As mentioned in Section 4.1 in the main text, our toy example ground truth is the function $f(\mathbf{x}) = \sin(10|\mathbf{x}|)$ with input in $\mathbb{R}^2$. This function was picked due to its extreme nonlinearity and difficulty to fit by standard neural networks. To increase the challenge even further, for training data, normal noise of size $\mathcal{N}(0,1)$ is added, while no noise is added to the test data. The "original dataset" consists of 50000 points with both dimensions between -1 and 1, while the "fine-tune dataset" consists of 50000 points with both dimensions between 0.8 and 1.5.

The model itself is a 3-layer MLP, consisting of weight layers [2, 500], [500, 500], and [500, 1]. LayerNorm Ba et al. (2016) is applied after every layer except for the last. RMSProp is used with default PyTorch hyperparameters ($\alpha = 0.99$, $\epsilon = 1e - 8$) and learning rate $1e - 2$ for fitting to the original distribution, with a learning rate decay of 0.9 every 500 steps. After the original distribution has been fit to, we fine-tune on the new distribution for 1500 steps at a learning rate of $5e - 4$, with a decay factor of 0.95 every 100 steps. The GOODFIT runs are run with $n_{\mathrm{ref}} = 1$.

### 3.2 LONG-TAILED IMAGE CLASSIFICATION

**Why is this challenging?** As previously discussed, we train a first-stage baseline on classes $\in [0, 89]$ on CIFAR100-LT. The second-stage long-tail fine-tuning involves training only on held-out classes $\in [90, 99]$. At first glance, this may appear an intuitive way to test our method. Consider the extreme case of $\beta = 10$. The first-stage training on classes $\in [0, 89]$ involves 19021 examples in the training set. The second fine-tuning step on classes $\in [90, 99]$ includes only 552 examples, which is $\sim 35\times$ fewer! For $\beta = 5$, this ratio is $\sim 22$ (23784 in first-stage vs 1072 in second-stage) and for $\beta = 2$, this ratio is $\sim 13$ (33453 in first-stage vs 2576 in second-stage). Due to this extreme imbalance, the fine-tuning step understandably results in catastrophically forgetting the baseline classes. Nonetheless, to demonstrate the ability of GOODFIT in mitigating catastrophic forgetting while showing superior long-tail performance, we choose these challenging settings.

**Implementation Details** For CIFAR100-LT experiments, we use GMLC Du et al. (2023) as our baseline. GMLC is a state-of-the-art long-tail classification method that achieves strong performance on CIFAR10-LT, CIFAR100-LT, and Imagenet-LT benchmarks. For each input batch, a local and global augmentation is applied. The global augmentation used is MixUp Zhang et al. (2017), which generates an image by mixing across two different classes. On the other hand, the local augmentation CutMix Yun et al. (2019) replaces a patch in the image with a region from another image. The embeddings from the encoder are fed through a projection head and the cosine similarity is maximized

between global and local image pairs. The encoder embeddings are also fed through a classification head which minimizes the supervised cross-entropy loss. The advantages of GMLC are that it performs a single-stage training and does not require negative samples for contrastive training. For further details, we refer the reader to Du et al. (2023).

For the first stage training, we use the SGD with a momentum of 0.9 and a weight decay of $5e - 3$. We train for 200 epochs with a batch size of 64 and a learning rate of $1e - 2$. For the second stage, we train the SGD counter-part for 40 epochs and GOODFIT for 60 epochs. For imbalance ratio $\beta = 2$ we use a learning rate of $1e - 3$, for $\beta = 5$ we use a learning rate of $5e - 4$, and for $\beta = 10$ we use a learning rate of $1e - 4$. For a fair comparison, we do not alter any other hyper-parameters during the fine-tuning stage. In all our experiments, we use the default loss weighting parameters i.e., a contrastive loss weight of 10 and a classification loss weight of 1.2. For GOODFIT, we use $n_{\text{ref}} = 1$ and $\lambda_{\text{ref}}$ to 1/1000 of the learning rate. All experiments are run on a single Tesla T4 GPU machine. The first-stage training requires 3 hours, while long-tail fine-tuning takes 10 minutes.

**Evaluation** As mentioned in the main paper, there is a steep trade-off between baseline class vs long-tail class performance. In order to evaluate both methods at discrete thresholds (Table 2 in the main paper), we fit a curve and evaluate this curve at the desired thresholds (in our case, 40%, 35%, and 30%).

### 3.3 LARGE-SCALE ROBOTICS MOTION PREDICTION

The motion prediction model follows a standard encoder-decoder transformer architecture, as in Nayakanti et al. (2022).

The encoder takes multi-modal inputs as the target agent's history, nearby agent histories, map information, and traffic light states. Each input modality is encoded by a separate MLP to an embedding with a dimesion of 64. The input embeddings are fused through concatenation as input tokens to a self-attention transformer. The encoder transformer includes 2 attention layers, 8 heads, 256 hidden dimensions, and 1024 feedforward dimensions. We add learned positional embeddings, initialized as a Gaussian vector with zero mean and standard deviation of 0.02, to each token.

The decoder is a cross-attention transformer that attends six learnable latent queries, initialized with zero mean and standard deviation of 0.02, to encoder embeddings. The decoder transformer includes 8 attention layers, 8 heads, 256 hidden dimensions, and 1024 feedforward dimensions. The output queries are mapped to a weighted set of six trajectory samples through an MLP. Each sample includes $(x, y)$ positions for the next 80 timesteps and a weight scalar.

The model is trained end-to-end by a smooth L1 loss on the trajectory predictions and a cross-entropy loss on the predicted weights. The AdamW optimizer is used with default PyTorch hyperparameters: learning rate = $1e - 3$, $\beta s = (0.9, 0.999)$, weight decay = $1e - 2$. The base model is trained on the WOMD training set for 60 epochs with a batch size of 256.

For car-to-car fine-tuning experiments, learning rate is dropped by a factor of 100 from the original and training is performed for only 1500 steps (because the original training run already converged, training for too long in this setting leads to overfitting). For car-to-pedestrian fine-tuning experiments, learning rate also is dropped by a factor of 100, but training is allowed to run for the same number of steps as the original model.

GOODFIT $n_{\text{ref}}$ is set to 3, and $\lambda_{\text{ref}}$ is set to 1/10 of the learning rate. We note that the value of $n_{\text{ref}}$ is quite high in this setting, but training is relatively fast and you can get better results fairly early on in training so the number of steps can be cut down considerably (see, for example, the curves in Figure 4).

## 4 LIMITATIONS

GOODFIT is designed exclusively for fine-tuning a pre-trained deep network and assumes access to a good initialization. Due to this assumption, GOODFIT cannot be used for training a neural network from scratch (i.e., randomly initialized weights) and will inexplicably perform poorly. This can be considered a limitation in our work. In other words, if one wishes to add a new head to an

existing model, the head would need to be pre-trained using some other optimizer from scratch before GOODFIT can be used.

Finally, since GOODFIT instantiates 2 optimizers (a main and a reference optimizer) as part of the update, it incurs a slightly increased memory footprint to keep track of optimizer parameters (depending on the optimizer of choice).