# OpenReview forum: "GOODFIT: A Deep Learning Optimizer Fine Tuned for Fine Tuning"
_ICLR.cc/2024/Conference — ICLR 2024 Conference Withdrawn Submission_

### Official Review · Reviewer_ZLp3 · 2023-10-31

**Soundness:** 2 fair
**Presentation:** 3 good
**Contribution:** 3 good
**Rating:** 5
**Confidence:** 3

**Summary:**

The paper introduces a new optimizer to finetune models that already converged on a pre-training corpus. The optimizer projects the gradients computed on the downstream task orthogonally to the direction moving away from the parameter state converged on the pre-training data. This strategy allows one to learn a new task without forgetting the old one.

**Strengths:**

Overall, the paper is well-written and easy to follow. It discusses an idea for optimizing pre-trained models, which is of the utmost importance for the field. This is especially important in the NLP domain, where the practitioners are fine-tuning foundational models pre-trained on large text corpora.

**Weaknesses:**

The experimental section needs to be revised. In particular, it should be extended to NLP, a domain in which the GOODFIT would be naturally applied.

**Applicability issue in LLMs**\
Methods for finetuning LLMs currently rely on low-rank approximations of the weight matrices (LoRA) or through adapter heads. These are usually randomly initialized or zero-initialized (see e.g. LoRA [1]) and trained on top/aside from the pre-trained model. GOODFIT can not be applied as-is to state-of-the-art models for NLP since it requires starting from a converged weight matrix. Further validation of GOODFIT in one of those cases is crucial for the contribution of this work to be applicable exactly where it is most needed: in the case of LLMs.

**Extension of the experimental validation**\
More generally, the experimental tests should be expanded. Currently, there are three tests: one on synthetic data, another on CIFAR100, and a third on the Waymo Open Motion Dataset.  These tests are insightful and important, but only one is on a large-scale dataset, and none of those is done on natural images or textual data. \
Since the cost for finetuning a model pre-trained on ImageNet or a large text corpus is relatively low, and the current method's impact could be significant in the NLP domain, I suggest adding experimental evidence in support of the GOODFIT performance on real-world image data and/or text.

**Discussion of the limitation in the main text**\
The authors should have included a more thorough discussion of the limitations of their findings in the main text. The supplementary material contains a short discussion in which the authors address some of those, including the “slightly increased memory footprint.”
These statements should be made more quantitative because they depend on the finetuning setup (full rank, LoRA, ...) and the type of architecture they are applied to (e.g. transformers vs CNNs).\
Discussing the limitations of the current work in the main text strengthens the results of the paper while hiding them in the supplementary material does not.

[1] Hu et al., *LoRA: Low-Rank Adaptation of Large Language Models*

**Questions:**

**Minor:**

1. **Correctness on old data**- *Proof sketch*: \
the claim seems to be valid for the initial step of the finetuning when the gradient of the $L_{old}$ parallel to $\Delta$. How can the argument be relevant for the following finetuning steps?

2. In the first line of the proof of the **stable points**: \
Should the statement -$\nabla_\theta L_{new}$ be $-\nabla_{\theta’} L_{new}$ instead? i.e. the gradient at the new $\theta’$ point?

3. The authors should add a link to the repo containing the code to reproduce the experiments of the paper. This is now a *de facto* mandatory requirement in this field.

---

### Official Review · Reviewer_BuWp · 2023-10-31

**Soundness:** 2 fair
**Presentation:** 3 good
**Contribution:** 2 fair
**Rating:** 3
**Confidence:** 3

**Summary:**

This paper proposes an optimizer for finetuning a well-trained model named GOODFIT. It takes advantage of the additional structure of a converged model and orthogonalization process to regularize the optimization process for better results. Reasonable results are reported.

**Strengths:**

1. The motivation is clear.
2. The proposed method is intuitive and makes sense to me.
3. The paper is well organized.

**Weaknesses:**

1. My primary concern is the insufficient empirical evaluation. Since the training dynamics of neural networks are still a black box, all new optimizers that claim to be effective must be thoroughly validated. However, in this paper, only results on toy datasets are reported, which cannot convince me at all.

2. The computational cost of GOODFIT is significant. Even with n_ref = 1, the cost is 2x as a regular optimizer, which may be unaffordable. And due to the extra costs, the baselines in the experiment part must also include finetuning results with 2x iterations, which are missing.

**Questions:**

I would suggest the authors include the following experiments.

1. Finetune real-world models on real-world datasets, for example, Swin/ViT/ConvNeXt on ADE20K/COCO. I would expect better performance than a trivial finetuning.

2. Finetune models that are considered to be difficult to finetune, e.g., CLIP-ViT on ImageNet. I would expect the finetuning accuracy to be higher and the original representation (e.g., zero-shot open-world recognition ability of CLIP) preserved, compared to a trivial finetuning.

3. (Optional, since it is too expensive) Finetune an LLM or multimodal LLM on some SFT datasets, since it is mentioned in the Introduction section.

---

### Official Review · Reviewer_aj9V · 2023-11-02

**Soundness:** 2 fair
**Presentation:** 3 good
**Contribution:** 2 fair
**Rating:** 3
**Confidence:** 4

**Summary:**

This paper introduces a tailored optimizer designed for fine-tuning on a novel task or dataset. It employs a temporal gradient orthogonalization process, which involves keep tracking record of a reference state and orthogonal projection of gradients onto it. This dynamic approach effectively maintains the model's proximity to the baseline model. The results of GOODFIT demonstrate consistent performance enhancements across various settings compared to both the full fine-tuning baseline and the fine-tuning head-only baseline."

**Strengths:**

The proposed method is simple and reasonable. It proposes to keep track of a “good” reference state and restores the functionality of this state by translating and orthogonalizing gradient updates with respect to this state. It also provides brief theoretical justification and detailed experimental analysis.

**Weaknesses:**

1): The paper could benefit from a more comprehensive discussion and comparison of training with regularization techniques. For instance, it should at least consider basic regularization methods like weight decay, which can significantly improve training efficiency with fewer hyperparameters.

2): The key point of the paper assumes that if the stored reference state corresponds to a good critical point of the original model, then $\Delta$ corresponds to a good gradient direction that restores the original critical point. However, as optimization progresses, there is no guarantee that $\Delta$ always corresponds to an optimal direction.

3): The paper lacks a rigorous theoretical proof related to generalization or convergence analysis. In the informal proof, what if the pretraining loss $L_{old}$ and fine-tuning loss $L_{new}$ are quite different? In such cases, the loss surface can become notably distinct.

4):  The paper does not provide a clear discussion of fine-tuning efficiency. Fine-tuning speed is at least 2$\times$ slower than vanilla fine-tuning, and it requires storing reference states in GPU memory during fine-tuning. This might be challenging to generalize to large foundation models with billions of parameters.

5):  Few ablation studies are included in the paper. For instance, there is no exploration of the effects of different configurations for the three main hyperparameters mentioned in Section 3.2.

6):  Both training and validation curves are expected to be included to demonstrate the regularization effect. As shown in Table 1, head only fine-tuning outperforms the full model fine-tuning, indicating that the model is overfitting and many regularization techniques might work, questioning the necessity of GOODFIT.

7):  The scalability and generalization capability of the proposed GOODFIT is not convincing. For example, how about using different optimizers, such as SGD variants, Adam, etc? Furthermore, experiments with large foundation models such as LLaMA and SAM would provide valuable insights.

8): The presentation can be improved. For example, the symbols in Figure 1 should align with the method descriptions.

**Questions:**

See weaknesses

---

### Official Review · Reviewer_vuSA · 2023-11-09

**Soundness:** 2 fair
**Presentation:** 2 fair
**Contribution:** 2 fair
**Rating:** 5
**Confidence:** 3

**Summary:**

The authors propose an optimization method  based on temporal gradient orthogonalization designed for finetuning. It is evaluated on CIFAR100-LT and Waymo Open Motion Dataset

**Strengths:**

- The objective of this work tackles a very important problem, the initial results are promising
- The method has some intuitive motivation and some theoretical results are provided

**Weaknesses:**

- The hyperparameters of the reference methods are important to fully evaluate the strengths of the work, they should be described in more detail
- The connection to LwF a method which uses distillation seems a bit vague and is relied on in several places
- Related Work: The authors set out to tackle a very widely studied problem of finetuning with a vast literature. I was surprised to not see much mention of meta-learning methods which have been extensively used for “finetuning”. What is the connection to  works such as
- Orthogonal Gradient Descent for Continual Learning
- Lookahead Optimizers
- Reptile and MAML
- Experiments - the experimental results are promising, but I would like to see better motivation for the datasets used. Transfer learning is widely studied and there are many standard datasets and fine tuning tasks results in the literature. The topic of this paper is a potentially high impact one but given the claims of providing a better generic finetuning method  I believe the experimental results need to be more comprehensive and include datasets like e.g. VTAB
- Theoretical analysis: some of the assumptions seem to undermine the interest of the results

**Questions:**

Would this method work well if the target data differed substantially from the source training model?